# A Remote Sensing-Based Analysis of the Impact of Syrian Crisis on Agricultural Land Abandonment in Yarmouk River Basin

**DOI:** 10.3390/s22103931

**Published:** 2022-05-23

**Authors:** Khaled Hazaymeh, Wahib Sahwan, Sattam Al Shogoor, Brigitta Schütt

**Affiliations:** 1Department of Geography, Yarmouk University, Irbid 21163, Jordan; khazaymeh@yu.edu.jo; 2Physical Geography, Institute of Geographical Sciences, Freie Universidad Berlin, Malteserstraße 74-100, 12449 Berlin, Germany; brigitta.schuett@fu-berlin.de; 3Department of Geography, Faculty of Social Sciences, Mutah University, AlKarak 61710, Jordan; sattam_1975@mutah.edu.jo

**Keywords:** Landsat, human conflict, NDVI, Syria, Jordan, GEE

## Abstract

In this study, we implemented a remote sensing-based approach for monitoring abandoned agricultural land in the Yarmouk River Basin (YRB) in Southern Syria and Northern Jordan during the Syrian crisis. A time series analysis for the Normalized Difference Vegetation Index (NDVI) and Normalized Difference Moisture Index (NDMI) was conducted using 1650 multi-temporal images from Landsat-5 and Landsat-8 between 1986 and 2021. We analyzed the agricultural phenological profiles and investigated the impact of the Syrian crisis on agricultural activities in YRB. The analysis was performed using JavaScript commands in Google Earth Engine. The results confirmed the impact of the Syrian crisis on agricultural land use. The phenological characteristics of NDVI and NDMI during the crisis (2013–2021) were compared to the phenological profiles for the period before the crisis (1986–2010). The NDVI and NDMI profiles had smooth, bell-shaped, and single beak NDVI and NDMI values during the period of crisis in comparison to those irregular phenological profiles for the period before the crisis or during the de-escalation/reconciliation period in the study area. The maximum average NDVI and NDMI values was found in March during the crisis, indicating the progress of natural vegetation and fallow land, while they fluctuated between March and April before the crisis or during the de-escalation/reconciliation period, indicating regular agricultural and cultivation practices.

## 1. Introduction

Abandoned agricultural land is defined as lands without plantation practices or grazing management such as plowing, planting, and intensive grazing for specific growing seasons [1,2,3]. It has been reported as one of the most common consequences of environmental changes and human conflicts [3,4]. It affects the vegetation dynamics, agricultural practices, field productivity, crop production, and perhaps food security in the long run. Therefore, an accurate characterization of abandoned agricultural land is needed. Consequently, the spatial and temporal patterns of abandoned agricultural land through the growing seasons and its related environmental factors have attracted increasing attention during the past several decades [5,6].

Accurate identification and characterization of abandoned agricultural land starts by firstly defining the growing season phenological metrics such as the start of season (SOS), end of season (EOS), and length of season (LOS) in the study area. In this context, researchers have used various methods to define growing season metrics such as in situ observations, model simulations, and satellite-based vegetation indices (VIs) [7,8,9]. In this regard, the last of those is the most used as it offers an opportunity to capture vegetation phenology across various spatial and temporal scales [10]. The identification of those metrics can be a good indicator for monitoring agricultural activities and management practices of agricultural lands. Moreover, they are very important for understanding the vegetation dynamics and their long-term characteristics which may change due to various environmental and human impacts on different spatial scales. Among the numerous VIs, the normalized difference vegetation index (NDVI) is the most used in retrieving the vegetation phenological metrics [11,12]. Additionally, the analysis of NDVI over continuous growing seasons in comparison to the long-term measures could be used for abandoned agricultural land mapping and monitoring. In this context, two approaches can be defined, namely the statistical and the phenological approaches

The statistical approach relied on analyzing the measures of the maximum, minimum, and average values of the NDVI spectral index. This approach provides a better understanding of the data than using the individual values, particularly when dealing with a huge number of datasets. Here, abandoned agricultural lands could maintain higher NDVI values for a longer period during the growing season than active agricultural lands and managed grasslands [11].

The phenological approach looks into the shapes and patterns of NDVI profiles in the growing seasons. This provides information about the agricultural activities within the area of interest. For instance, the phenological profiles of abandoned agricultural land would be characterized by regular, bell-shaped, and less temporal fluctuations during the growing season due to the succession of weeds, grasses, and eventually shrubs or trees [12,13,14,15]. While the managed agricultural lands would be characterized by more irregular temporal shapes with one or more narrow peaks within the growing season. Such management practices (e.g., plowing, mowing, and grazing) lead to abrupt changes in those NDVI temporal profiles.

Mapping and monitoring of abandoned agricultural land have been conducted using various satellite systems including the Moderate Resolution Imaging Spectroradiometer (MODIS), Visible Infrared Imaging Radiometer Suite (VIIRS), Satellite Pour l’Observation de la Terre (SPOT), Sentinel-2, and Landsat [3,11,16,17,18,19]. However, the success of any approach depends on the resolution of remote sensing data and the used algorithms for characterization of wild vegetation from active crops [20,21]. Among the different satellite systems, Landsat images are the most used as they provide consistent spatial and temporal information since 1972 at proper spatial (i.e., 30 m) and temporal (i.e., 16 days) resolutions. This allows for analyzing the terrestrial vegetation dynamic including abandoned agricultural land at regional and local scales [22]. Researchers have suggested the use of three Landsat images per year, both pre- and post-abandonment, to achieve abandoned agricultural land mapping with accuracy of up to 80% if sufficient cloud-free images were available [3]. Yin et al. [23] employed time-series Landsat images for mapping abandoned agricultural land in the Czech Republic and linked them to the Russian–Chechen War hotspots in the Caucasus region. Gbanie et al. [24] performed trajectory analysis of Landsat and SPOT images in the Sierra Leone during the war (1976–2000) and after the war (2003–2011). They found that urban and peri-urban agriculture became a major livelihood activity for displaced persons for food production. Baumann et al. [25] used a combination of linear regression models and hierarchy for mapping the spatial determinants in the post-socialist farmland abandonment in Ukraine using Landsat images from 1986 to 2008. They also found that farmland abandonment was widespread in the study region at abandonment rates of up to 56%. Witmer [26] investigated the war-induced abandoned agricultural land in northeast Bosnia using Landsat-5 images and identified abandoned agricultural land with an overall accuracy of 82.5%. Similar studies have been conducted in other conflict areas in eastern Europe [27]; South Sudan [28]; Lebanon [29]; and Iraq [30,31]. All previous studies have used either multi-temporal satellite images for detecting phenological dynamics throughout short periods, or single images for mapping vegetation succession that invades abandoned croplands and fields.

During the Syrian crisis, abandoned agricultural land has been documented in several parts of Syria with varying negative impacts [32]. The massive insecurity situation along with the border areas between Syria and Jordan due to the crisis caused degradation in water resources, agricultural activities, and exportation of agricultural products [33]. The area in southern Syria and northern Jordan is the main contributor to rainfed agriculture and comes in small, fragmented ownership size agricultural lands. This area includes the Yarmouk River Basin (YRB), which is one of the most important river basins in the region. This study aims to evaluate the impact of the Syrian crisis on agricultural land abandonment in the YRB during the period from 2011 to 2021 through evaluating the phenological characteristics of agricultural land using all the historical Landsat data starting from 1986.

## 2. Materials and Methods

### 2.1. Study Region

The Yarmouk River Basin (YRB) is a transboundary basin shared between Jordan and Syria (Figure 1). The total area of the basin is 7242 km^2^. The majority of the basin (i.e., 5818 km^2^~80.3%) is located in Syria [34]. The elevation varies from 1858 m a.m.s.l. at Jabal Al-Arab, then it fluctuates between 500 m and 650 m a.m.s.l. in the plain fields in the middle area and reaches its lowest point at 212 m a.m.s.l. in its outlet in the Jordan River. YRB has undergone increasing development in urban, industrial, and agricultural activities and plays a significant role in the socio-economic development of both countries. The basin is characterized by a semiarid Mediterranean climate in the west and an arid climate in the east. The winter season starts in November and ends in early May. The average annual rainfall varies from the east (~150 mm) to the west (~600 mm). The average annual minimum and maximum temperature values range between 9 °C and 24 °C in the east and from 12 °C to 23 °C in the west, respectively. Most of the basin is covered by agricultural land including rainfed crops such as cereals, olive trees, and vegetables in the west, while the eastern parts extend to the low rainfall zone where irrigation occurs using groundwater [35].

### 2.2. Data Collection

#### 2.2.1. Satellite Data

In this study, time-series surface reflectance images for 35 years (i.e., 1986–2021) acquired by Landsat-5 and Landsat-8 satellites were used. The dataset was freely obtained from the United States Geological Survey (http://lpdaac.usgs.gov accessed on 15 November 2021) and processed using JavaScript commands in Google Earth Engine (GEE). As a matter of fact, GEE is indeed a web-based open-source processing platform that combines a multi-petabyte catalog of satellite imagery and geospatial datasets with planetary-scale analysis capabilities. It uses Java scripts for downloading satellite data, detecting landscape changes, classifying land cover types, mapping trends, and quantifying differences on the Earth’s surface, and has been recently used for various remote sensing applications [36,37]. The dataset was collected from LANDSAT/LT05/C01/T1_SR and LANDSAT/LC08/C01/T1_SR for Landsat-5 and Landsat-8, respectively. These datasets are geometrically registered and processed to orthorectified surface reflectance. They were atmospherically corrected using the LEDAPS algorithm for Landsat-5 and the LaSRC algorithm for Landsat 8 and include a cloud, shadow, water, and snow mask produced using CFMASK, as well as a per-pixel saturation mask [38].

We selected the images with less than 10% cloud contamination from the entire image collection of Landsat-5 from 1986 to 2010 and Landsat-8 from 2013 to 2021. The total number of processed images was 1155 and 495 for the two image collections, respectively. All the images were atmospherically corrected and georeferenced to UTM Zone 36 N. We excluded the years 2011 and 2012 due to the data unavailability.

To define the extent of potentially abandoned agricultural land in the study area, we firstly used Copernicus Global Land Service (CGLS collection 3) land cover map for the year 2019 as a background base-map to define the agricultural land within the study area. The CGLS is a global land cover map of 100 m spatial resolution provided for the period 2015–2019 over the entire globe; it is derived from the PROBA-V 100 m time-series with an accuracy of 80% at Level 1 for all years [39]. In this study, we focused on agricultural land types among the other CGLS 23 classes; therefore, we generated a mask to exclude forests, urban areas, and barren land. Accordingly, all types of agriculture were kept (i.e., rainfed or irrigated agriculture, grassland, annual crops associated with permanent crops and complex cultivation patterns, mixed agriculture/natural vegetation, and mixture of natural vegetation, herbaceous, shrub, and trees). These subclasses were considered to identify the potential unmanaged areas in the study area which could be active or abandoned agricultural lands during the study period (see Figure 1).

#### 2.2.2. War and Internally Displaced Persons (IDPs) Data

The areas of YRB have witnessed armed activities starting from August 2011. Since then, the security situation in this part of Syria has been precarious. It has been the scene of intensive armed battles and interchangeable control authorities between the conflict parties, which were reflected in the forced movement of people. For instance, the governorates of Dara’a, Quneitra, As-Sweida, and Rural Damascus which consist the Syrian part of YRB witnessed massive internal movements of people within or between the other regions of Syria, and external movements to Jordan due to the armed conflict. By 2016, the total number of registered Syrian refugees in Jordan reached 953,289 persons residing either in refugee camps or in the cities, towns, and villages. Approximately 47.6% of them (i.e., 453,486 persons) reside in Irbid and Mafraq governorates which consist the Jordanian part of YRB [40]. The IDPs have reportedly faced protracted internal displacement since the closing of border crossings between Syria and Jordan in June 2016, which has prevented further movement into Jordan. By December 2018, the United Nations Office for the Coordination of Human Affairs (UN-OCHA) estimated the total number of IDPs in these four governorates at 997,215 persons. However, during the de-escalation/reconciliation period (2018–2021) the UN-OCHA documented 903,370 spontaneous IDP returnees in these governorates, where the major returns occurred between August 2018 and March 2019. Figure 2 shows a summary of the total IDP spontaneous returnees in these governorates during that period [41].

#### 2.2.3. Rainfall Data

The monthly average rainfall data (Table 1) between 1986 and 2021 obtained from three agro-climatic stations (i.e., As-Sweida, Dara’a, and Quneitra) within the Syrian part of YRB were used for evaluating the potential effects of rainfall on agricultural land extent, crop yield and, agricultural production during the years of crisis. The agricultural data during 2011–2021 were obtained from the Syrian Ministry of Agriculture. It contains statistics on the total cultivated area (ha), yield (ton/ha), and production (ton) of the major field crops (i.e., durum wheat, barley, lentils, peas, and green beans) in Dara’a, Quneitra, As-Sweida, and Rural Damascus which consist the area of YRB (note that the data for 2021 is not available). These data were used to analyze the agricultural activities during different growing seasons before, during, and within the calm years of crisis in YRB.

### 2.3. Mapping Abandoned Agricultural Land in YRB

#### 2.3.1. Defining the Growing Season

In this study, we used the normalized difference vegetation index (NDVI; Equation (1)) and normalized difference moisture index (NDMI; Equation (2)) to identify the growing season within the study area as they reflect the greenness and wetness conditions of vegetation and are frequently used in vegetation monitoring [42].
(1)$$NDVI=\frac{NIR\;-\;Red}{NIR+Red}\;$$
(2)$$NDMI=\frac{NIR\;-\;SWIR}{NIR+SWIR}$$
where *NIR*, *Red*, and *SWIR* are near infrared, red, and shortwave infrared spectral bands of Landsat 5 and Landsat 8. Here, we calculated the long-term monthly average NDVI and NDMI values for the whole study area from 1986 to 2021 by implementing Java scripts on GEE. We used the average monthly NDVI values to define the growing season metrics, the start of season (SOS), length of season (LOS), and end of season (EOS), taking into consideration that the NDVI values tend to increase starting from the SOS till they reach a peak value, then they start to decrease toward the end of the EOS [43]. We calculated the long-term averaged distribution and spatial trends of these growing season metrics at the pixel level in the whole study area during 1986–2021.

#### 2.3.2. Identifying Abandoned Agricultural Land Characteristics in the Yarmouk River Basin

We performed two approaches for identifying the abandoned agricultural lands in YRB: (i) an analysis of the variation in the maximum and average NDVI and NDMI values for the growing season (i.e., February to May in our study area) during the period of observation (i.e., 1986–2021) bearing in mind that these NDVI and NDMI values would vary between abandoned and active agricultural lands; (ii) an analysis of the long-term phenological profiles of NDVI and NDMI values of the growing seasons between 1986–2020, taking in consideration that these NDVI and NDMI temporal profiles could provide information about the agricultural activities within YRB during the period of interest and especially during the conflict period in Syria.

We also performed a country-specific analysis of YRB in Jordan and Syria. As such, we divided the basin into two parts (Figure 1) following the main channel of the Yarmouk River which also represents the political divide boundary between the two countries. This was done to better understand the effects of the Syrian crisis on agricultural lands and agricultural activities between both sides of the basin. Accordingly, we used 150 on-screen digitized sample points within each country-specific basin (i.e., 111 in the Syrian part and 39 in the Jordanian part) around the towns and villages to characterize the agricultural land in these areas in relation to the conflict actions, human displacement, and agricultural activities.

#### 2.3.3. Evaluating the Effects of the Syrian Crisis on Agricultural Land Abandonment in YRB

The monthly average rainfall data, the data of internally displaced persons (IDPs) and spontaneous returnees, and the agricultural data were used to evaluate the remote-sensing-based results of mapping abandoned agricultural land in YRB. For instance, we examined whether the rainfall and/or the IDPs/returnees had influenced the agricultural activities, cultivation, yield, and crop production within the YRB, especially in its Syrian part, during the period of the crisis between 2011 and 2021 or not.

## 3. Results

### 3.1. Defining the Metrics of the Growing Season in the YRB

The analysis of the long-term monthly average NDVI values for the agricultural land-use type during the period 1986–2021 shows that only one growing season exists in the study area (Figure 3). The phenological curve of NDVI starts mainly in February (i.e., SOS) when crops start to germinate, and ends in May (i.e., EOS) when crops reach their maturity. The highest NDVI values were found in March and April. The length of the growing season (LOS) was found to be approximately four months. We also calculated the monthly average NDVI for each year between 1986 and 2021 and found negligible differences in NDVI values of SOS, peak, and EOS between the growing seasons which might be related to variations in agricultural practices, seasonal climate conditions, and crop types. Here, the SOS or EOS were defined as the month of the year at which the left or right edges of the monthly averaged NDVI profile increase or decrease, respectively, to 20% of the highest amplitude in the NDVI profile of a growing season, and the LOS is the time in months between the start and the end of the growing season [11].

### 3.2. Analysis of the NDVI Phenological Profiles in the YRB

The spatiotemporal phenological profile of agricultural land-use type is shown in Figure 4. It represents the monthly average NDVI and NDMI values of the growing seasons (i.e., February to May) for 35 years from 1986 to 2021 using Landsat-5 and Landsat-8 data. Taking in consideration the definition of abandoned agricultural land and the characteristics of monthly average NDVI and NDMI profiles, three main features can be distinguished in Figure 4: (i) phenological profiles that have irregular values that fluctuated with various peaks between 1986 and 2010, where the maximum average NDVI and NDMI values often altered substantially between March and April; (ii) phenological profiles with smooth, bell-shaped, and single-peak NDVI and NDMI values between 2013 and 2018, where the maximum average NDVI and NDMI values occurred mainly in March, indicating the progress of natural vegetation and fallow land; and (iii) a second cycle of irregular NDVI and NDMI profiles during 2019–2021 with noticeable higher NDVI and NDMI values which might be related to the recultivation process and agricultural activities within the study area during that calm period of the crisis as the conflict parties signed a matter of de-escalation/reconciliation with no military actions in late 2018 in this part of Syria.

The analysis of NDVI and NDMI profiles at the country-specific parts of the YRB is shown in Figure 5. Figure 5a shows the NDVI and NDMI phenological profiles in the Jordanian part of YRB. It shows that the NDVI and NDMI phenological profiles had irregular fluctuated shapes and values in all growing seasons between 1986 and 2021. This can be connected to the active agricultural practices in the Jordanian part of the basin during that period, including the period of crisis between 2012 and 2021. In the Syrian side of the basin, the spatiotemporal NDVI and NDMI phenological profiles showed smooth, bell-shaped, and single-peak NDVI and NDMI values between 2013 and 2018 with an exception in the year 2015 (Figure 5b). Some other discrete growing seasons (i.e., 1987, 1990, 1993, and 1995) showed similar profile characteristics which might be due to other factors such as climatic and cultivation practices. The peak values of NDVI and NDMI in both parts of the basin were approximately close as they almost have similar climate conditions. However, the maximum average NDVI and NDMI values in the Jordanian part of YRB altered substantially between March and April in all growing seasons between 1986 and 2021 (Figure 6a). Hence, during the crisis period, many Syrian farmers, especially from the study area, moved to Jordan and contributed to the agricultural activities and production in the Jordanian part of the basin [32]. In the Syrian part of YRB, the maximum average NDVI and NDMI values were mainly found in March during the crisis years except for 2019, 2020, and almost 2021 (Figure 6b). This could confirm the existence of abandoned agriculture during the crisis years 2013–2018 due to the war activities when compared to the maximum NDVI and NDMI profiles before the crisis (1986–2010) and during the de-escalation/reconciliation period between 2019 and 2021.

### 3.3. Analysis of Agricultural Activities in YRB during the Syrian Crisis in Relation to Rainfall and IDPs/Spontaneous Returnees

The analysis of rainfall data showed that the rainfall was within its normal annual range during the study period, including the active, abandoned, and recultivated periods. The long-term average rainfall amount during the period 1986–2021 was 304.7 mm in As-Sweida, 247.2 mm in Dara’a, and 612.5 mm in Quneitra. The monthly rainfall in these stations in the years of the crisis 2011–2021 was found within its normal levels where there were no extremely high or low rainfall records during the crisis’s calm years in 2019, 2020, and 2021, as shown in Figure 7.

Figure 8 show the results of analyzing the statistics of agricultural data. Figure 8a shows that the total cultivated area was approximately equal in 2011 and 2012; then it started to reduce starting from 2013 until 2018. After that, the cultivated area was occasionally increased to reach in total an area greater than that in 2011 (i.e., before the war). Similar findings were observed in the crop yield (Figure 8b) and production (Figure 8c). These figures ensured the relationship between these three agricultural measurements, such as an increase in the total cultivated area and yield would reflect an increase in the total production. However, this was not the case in 2016, which showed a lower production value compared to the other years; hence, this might be related to a statistical mistake or missing data in that year.

## 4. Discussion

In this study, we provided the first consistent spatiotemporal analysis of abandoned agricultural land across the YRB in Syria and Jordan in relation to the Syrian crisis. The implementation of Landsat-based NDVI and NDMI data for identifying vegetation phenology successfully identified the phenological characteristics of the growing season in YRB. The statistics of NDVI and NDMI and their spatiotemporal profiles were generated using Landsat-5 TM and Landsat-8 OLI data for the study area. The NDVI and NDMI values were higher in the case of Landsat-8 data in all growing seasons. However, this might be related to the configuration variations between the two satellite sensors in terms of their spectral bandwidth and radiometric resolutions; similar findings were reported in previous work in different studies [44,45]. On the other hand, the spatiotemporal profiles of NDVI and NDMI showed similar patterns during the phenological cycle between the two satellite sensors. This might be related to the combined effect of leaf pigments, leaf structure, and leaf water content that gives all healthy green vegetation canopies their common reflectance properties such as low reflectance in red and SWIR spectral bands and high reflectance in NIR. The shape of the temporal profile of NDVI and NDMI from the TM and OLI sensors was used to successfully identify active/abandoned agriculture in the study area as they showed similar shapes and properties. Similar findings were observed using various satellite sensors in different studies and regions [14,23,26,27,28,29,30,31,43]. In identifying whether an abandonment case exists or does not, we investigated both the shapes and the values of the NDVI and NDMI phonological profiles for more than two successive growing seasons rather than looking at single-year comparisons. Factors such as weather conditions would be other factors affecting the agricultural status in a single growing season. For example, the growing season of 1995 had similar phonological shapes and values to that of 2014; however, it was followed by a normal NDVI and NDMI status in the following growing season. The growing season of 2014—for instance—was located within a successive period of more than two years that had similar phonological NDVI and NDMI profiles and values.

The implemented method in this study provided the information about the timing of land abandonment/recultivation occurrences which would be an indicator of the status of agricultural land management practices in the study area during the years of crisis. For instance, the recultivation process would be identified by investigating that in comparison to the cultivated area, yield, and production values. Those results could be valuable for land cover mapping and for assessing the land cover trajectories and identifying the agricultural abandonment years, as well as the periods when lands were recultivated. In addition, our analysis was suitable for assessing land abandonment at the regional/watershed level in relation to human conflicts. The analysis of the rainfall data confirmed that the climate conditions did not trigger the agricultural or the abandonment/recultivation activities in the study area during 2019–2021 as the total rainfall was within its long-term normal average. Thus, the possible effect of rainfall conditions on increasing the agricultural activities and perhaps the NDVI values and its temporal profiles in those years might be negligible.

The maps of IDP spontaneous returnees also supported the results of the remote-sensing-based NDVI and NDMI profiles. For instance, the study area contains major cities, towns, and villages where agriculture is the main economic activity for residents. These areas witnessed intensive airstrikes and ground armed battles between the conflict parties. Meanwhile, they witnessed different changeable control authorities between the government, the opposition, and other local armed groups during 2011–2021. UNOCH [37] reported that before August 2018 the subsequent shifts in the security situation generated multiple repeated displacements as the internally displaced persons (IDPs) left their place of displacement to return home or fled again when hostilities resumed, or kept moving onwards as hostilities unfolded. After that, due to the apparent ceasing of armed conflict and the reconciliations with local tribes in southern and southwestern Syria, in addition to the stable authority of the Syrian government, many people tended to return to their towns and villages. This situation leads to efforts by the people to recultivate their lands, which was obviously reflected in the NDVI and NDMI phenological profiles in 2019–2021 and gave them the irregular fluctuating shapes and various peak NDVI and NDMI values. This would be an obvious indicator of agricultural activities and management practices within the agricultural land in the study area during that period. Such fluctuations would correspond to the natural variation of the crop types, phenology, irrigation, soil conditions, agricultural and grazing practices and management, and climate conditions, including precipitation and temperature. The analysis of the agricultural statistics also confirmed our findings regarding the agricultural land abandonment/recultivation during the war years. The variation between the total cultivated area, yield, and production before and during the crisis years within the governorates of Dara’a, Quneitra, As-Sweida, and Rural Damascus coincided with the movement of people within these areas. These measurements increased or decreased consequently with the extent of the war and armed activities or the relatively calmer times which witnessed the return of the IDPs to their origin towns and villages within the study area.

## 5. Conclusions

Remote sensing data and techniques provide key information needed to support agricultural land monitoring, including agricultural land abandonment. The analysis focused on agricultural land in the Yarmouk River Basin in northern Jordan and southern Syria during 1986–2021. The operative and efficient acquisition of information from remote sensing often helps in these studies as such data are acquired at regular time intervals and various geographic scales, which makes it possible to monitor the development of agricultural land in deferent land sizes. Here, we tested and proved that Landsat-5 and Landsat-8 time series data are well suited for monitoring abandoned agricultural land during the conflict period in Syria using NDVI- and NDMI-based phenological analysis. The analysis confirmed the common belief about agricultural land abandonment in areas under conflict. The results pointed to multiple trajectories regarding agricultural land cultivation in the study area: (i) active agricultural period between 1986 and 2010, (ii) agricultural land abandonment in the conflict period between 2011 and 2019; and (iii) recultivation and active agriculture in the period 2020–2021. Our study highlighted the need to understand the diverse conflict-related changes to agricultural land for the purpose of agricultural management and rehabilitation in YRB. The results were evaluated against information from the Syrian Ministry of Agriculture for Dara’a, Quneitra, As-Sweida, and Rural Damascus governorates which consist the Syrian part of YRB. These included (i) the monthly historical rainfall data between 1986 and 2021, (ii) the total cultivated area, yield, and production of field crops during 2011–2020, and (iii) the internally displaced person (IDP) spontaneous returnees from 2018 to 2021. This evaluation confirmed the remote-sensing-based results. However, it is worthwhile to consider the other factors that may affect the agricultural land abandonment when implementing the proposed method in other study areas and environments. Meanwhile, the implementation of the method would be valuable for monitoring and evaluating the changes in agricultural crop types and cultivation patterns in the study area, which would be considerable for further investigation in the near future.

## Figures and Tables

**Figure 1 sensors-22-03931-f001:**
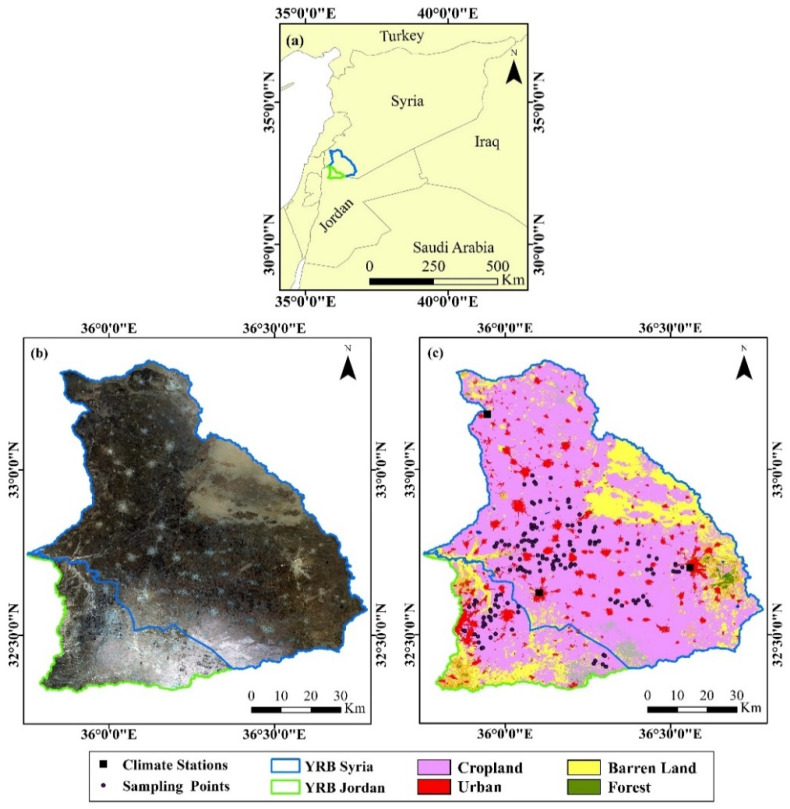
(**a**) Location map of Yarmouk River Basin (YRB) in Northern Jordan and Southern Syria, (**b**) a true Landsat-8 color image for 2021, and (**c**) land cover Map for YRB (CGLS, 2019).

**Figure 2 sensors-22-03931-f002:**
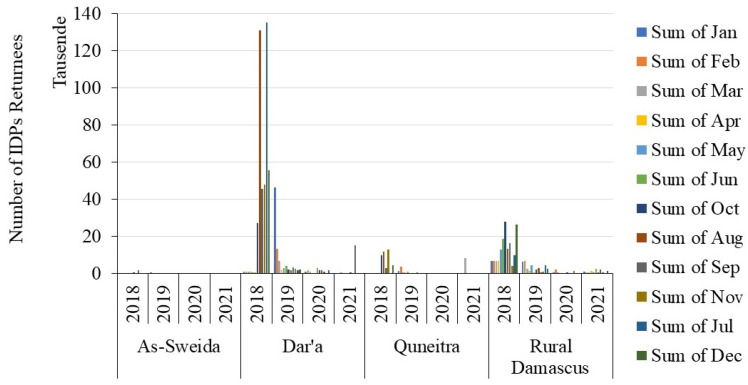
Summary of the total IDP spontaneous returnees in Dara’a, Quneitra, As-Sweida, and Rural Damascus governorates which mainly consist the Syrian part of YRB.

**Figure 3 sensors-22-03931-f003:**
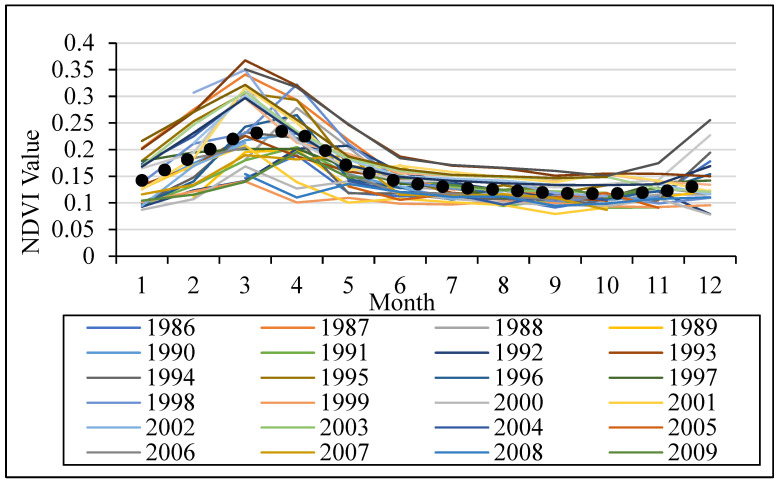
The monthly average NDVI profile for the years 1986–2021 for the agricultural land-use type in the Yarmouk River Basin (YRB), and the monthly long-term average NDVI between 1986 and 2021 (MLTA).

**Figure 4 sensors-22-03931-f004:**
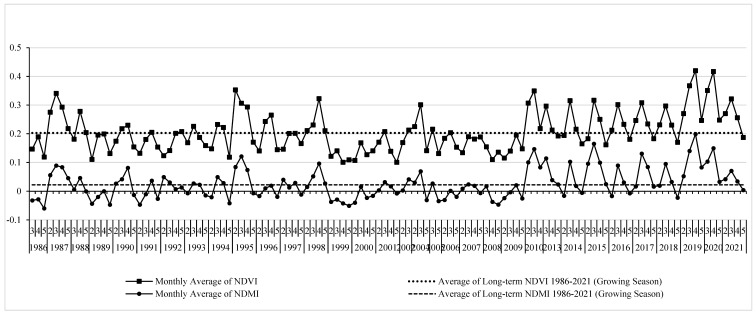
The spatiotemporal phenological profile of the agricultural land-use type using monthly average NDVI and NDMI values of the growing seasons (i.e., February to May) for the period 1986–2021 as extracted from Landsat-5 and Landsat-8 data in the Yarmouk River Basin (YRB).

**Figure 5 sensors-22-03931-f005:**
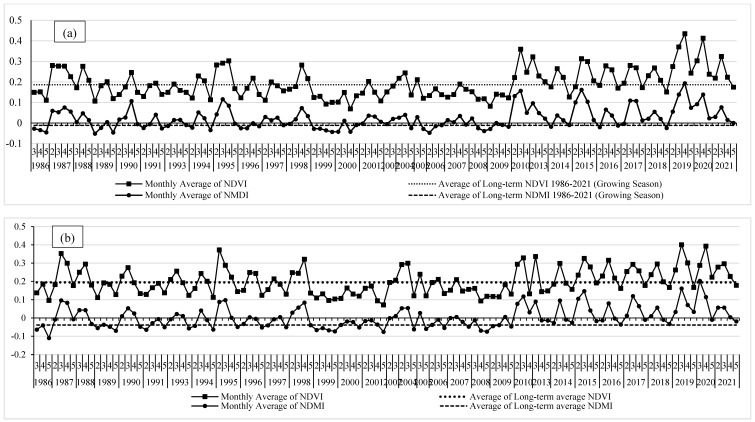
Spatiotemporal phenological profiles of the monthly average NDVI and NDMI values in (**a**) Jordanian and (**b**) Syrian parts of YRB during all growing seasons between 1986 and 2021.

**Figure 6 sensors-22-03931-f006:**
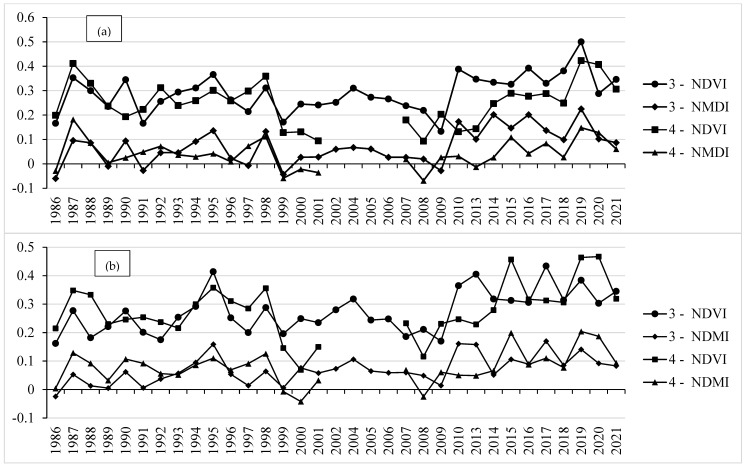
Spatiotemporal phenological profiles of the maximum NDVI and NDMI values in March and April during all growing seasons between 1986 and 2021 in (**a**) Jordanian and (**b**) Syrian parts of YRB.

**Figure 7 sensors-22-03931-f007:**
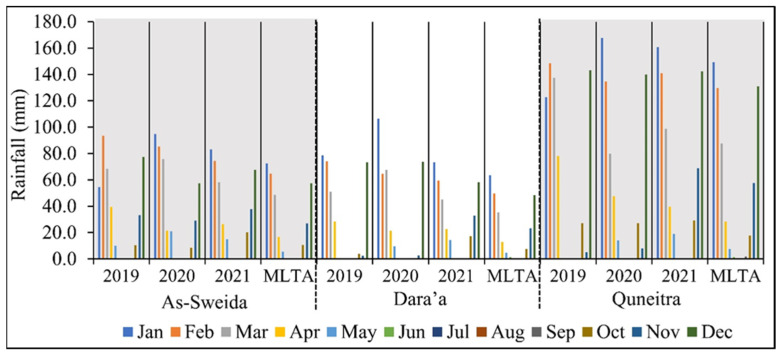
Statistics of the rainfall in three agroclimatic stations in the Syrian part of the Yarmouk River Basin during 2019–2021. MLTA is the monthly long-term average during 1986–2021.

**Figure 8 sensors-22-03931-f008:**
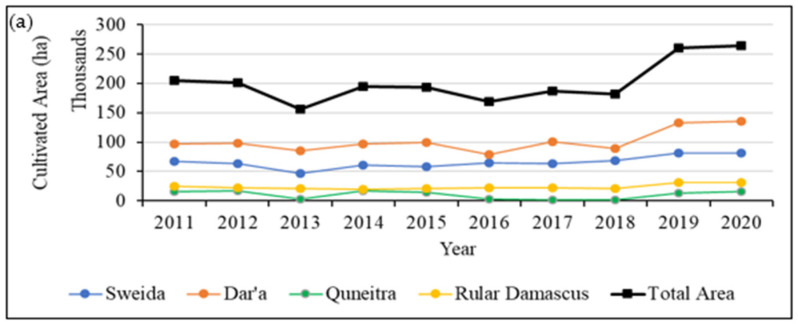
Statistics of (**a**) the total cultivated area, (**b**) crop yield, and (**c**) agricultural production in Dara’a, Quneitra, As-Sweida, Quneitra, and Rural Damascus governorates in the Syrian part of YRB (Data Source: Syrian Ministry of Agriculture).

**Table 1 sensors-22-03931-t001:** Statistics of the rainfall in three agroclimatic stations in the Syrian part of Yarmouk River Basin (YRB) during the years of the crisis 2019–2021.

	As-Sweida				Dara’a				Quneitra			
	2019	2020	2021	Monthly LTA *	2019	2020	2021	Monthly LTA	2019	2020	2021	Monthly LTA
January	54.5	94.7	83.1	72.5	78.5	106.5	73.2	63.5	122.7	167.5	160.5	149.2
February	93.5	85.3	74.4	64.7	74.0	64.6	59.5	49.8	148.5	134.7	140.9	129.6
March	68.5	75.7	58.2	48.5	51.0	67.7	45.0	35.3	137.5	79.7	98.8	87.5
Aprile	39.5	21.4	26.4	16.7	28.5	21.3	22.6	12.9	78.0	47.7	39.6	28.3
May	10.1	20.8	15.0	5.6	0.0	9.6	14.3	4.6	0.0	14.0	18.9	7.6
June	0.0	0.0	0.0	0.0	0.0	0.0	0.0	1.3	0.0	0.0	0.0	1.5
July	0.0	0.0	0.0	0.0	0.0	0.0	0.0	0.0	0.0	0.0	0.0	0.6
August	0.0	0.0	0.0	0.0	0.0	0.0	0.0	0.0	0.0	0.0	0.0	0.0
September	0.0	0.0	0.0	0.0	0.0	0.0	0.0	0.6	0.0	0.0	0.0	1.7
October	10.5	8.5	20.2	10.6	3.8	0.6	17.2	7.5	27.0	27.0	29.2	17.9
Nov	33.1	29.0	37.7	26.9	2.4	2.5	32.9	23.2	5.0	8.0	68.8	57.5
December	77.5	57.5	67.7	57.5	73.3	73.6	58.1	48.4	243.0	240.0	142.3	131.0
Yearly SUM	387.1	392.9	382.8		311.5	346.4	322.8		861.7	838.6	699.0	
LTA (1986–2021) is 304.7 mm					LTA (1986–2021) is 247.2 mm				LTA (1986–2021) is 612.5 mm			

* LTA is the long-term average rainfall during 1986–2021 (Source: Syrian Ministry of Agriculture).

## Data Availability

The data presented in this study are available on request from the first author.

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
