# Peer review of "A Remote Sensing-Based Analysis of the Impact of Syrian Crisis on Agricultural Land Abandonment in Yarmouk River Basin"

_sensors, 2022, doi:10.3390/s22103931_

Round 1

Reviewer 1 Report

This paper used Landsat-5 and Landsat-8 time series data between 1986-2021 to monitor abandoned agriculture land during the conflict period in Yarmouk River Basin (YRB) in Southern Syria and Northern Jordan by NDVI-based phenological analysis. The results showed the impact of the Syrian crisis on agriculture land abandonment in areas under conflict situation .The goals and the analyses are well described by the authors.
Detailed comments are as follows.
1.As the Landsat-5 and Landsat-8 time series data were processed to obtain the NDVI time series in GEE, the key steps of remote sensing data processing should be described in more detail.
2. Besides NDVI, the author should explained why not choose other vegetation indices for comparison and cross validation in the paper, as various factors can influence the VI values. 
3. The figure of spatial monthly average distribution of NDVI  during the period 1986-2021 in the study area could be added to demonstrate the seasonal dynamics.
4.Figure 8: please explain the correlation between the cultivated area,  crops yield and production, and why the lowest value of production occurred in 2016.

Author Response

Responses to reviewer # 1 are highlighted in green and the yellow ones are responses to both reviewers.

Reviewer 2 Report

This study examines the impact of the Syrian crisis on agricultural land abandonment in Yarmouk River basin, which covers parts of Jordan and Syria, using a time-series of 35 years of Landsat-derived NDVI. Generally, the manuscript was quite well written and easy to follow, and this is a quite interesting study. But I have some concerns as below:

When looking at the NDVI profiles, I am not convinced that we can identify the abandoned period (2013-2021). For example, in Figure 4, the monthly average NDVI profiles of 2014 and 1998 are quite similar. So how did you know 2014 was abandoned?

1986-2012 was defined as a normal cultivation period while NDVI values during this period were often lower than during 2013-2019 (Figure 4) => was your observation that crop NDVI value is normally lower than NDVI of natural vegetation? – If so, the values during 2020-2021 are even higher than that during 2012-2019 => how did you know these high values indicate the re-cultivation process?

This analysis was purely based on Landsat NDVI and I would think the results do not really draw your conclusions. Maybe we need ground truth data to verify the NDVI profiles, or we need further remote sensing metrics to support the findings.

Figure 1: the colour of sampling points should be changed. It quite hard to distinguish them from the climate stations

Author Response

Responses to reviewer # 2 are highlighted in blue and the yellow ones are responses to both reviewers.
